# Time to Micromanage the Pathogen-Host-Vector Interface: Considerations for Vaccine Development

**DOI:** 10.3390/vaccines7010010

**Published:** 2019-01-21

**Authors:** Jessica E. Manning, Tineke Cantaert

**Affiliations:** 1Laboratory of Malaria and Vector Research, National Institute of Allergy and Infectious Diseases, National Institutes of Health, Phnom Penh 12201, Cambodia; 2Immunology Unit, Institut Pasteur du Cambodge, Institut Pasteur International Network, Phnom Penh 12201, Cambodia; tcantaert@pasteur-kh.org

**Keywords:** vector-borne disease, saliva, tissue-resident memory cells, mosquito, tick, sandfly

## Abstract

The current increase in vector-borne disease worldwide necessitates novel approaches to vaccine development targeted to pathogens delivered by blood-feeding arthropod vectors into the host skin. A concept that is gaining traction in recent years is the contribution of the vector or vector-derived components, like salivary proteins, to host-pathogen interactions. Indeed, the triad of vector-host-pathogen interactions in the skin microenvironment can influence host innate and adaptive responses alike, providing an advantage to the pathogen to establish infection. A better understanding of this “bite site” microenvironment, along with how host and vector local microbiomes immunomodulate responses to pathogens, is required for future vaccines for vector-borne diseases. Microneedle administration of such vaccines may more closely mimic vector deposition of pathogen and saliva into the skin with the added benefit of near painless vaccine delivery. Focusing on the ‘micro’–from microenvironments to microbiomes to microneedles–may yield an improved generation of vector-borne disease vaccines in today’s increasingly complex world.

## 1. Introduction

Responsible for nearly one million deaths per year [1], vector-borne diseases are on the rise [2]. Vector-borne disease occurs when a pathogen is transmitted by the infected bite of a blood-feeding arthropod such as a mosquito, fly, mite, flea or tick. Effective vaccines are needed to counter this global public health threat. At present, research efforts are mostly focused on the pathogen-host interactions without acknowledging the significant contribution of vector-derived products to disease development. Yet, as early as the 1940s, physicians observed humans’ variable clinical responses to mosquito saliva and surmised that it may contain molecules with immunomodulatory capabilities [3]. However, not until the 1970s did scientists recognize the potential role of vector saliva in pathogen transmission [4]. Now, some attribute the complications of vector-borne disease vaccine development to the pleiotropic effects of vector saliva on both host and pathogen [5]. Pathogen-host interactions are complex in any disease setting, but become more nuanced with the addition of the vector, a previously underappreciated disease determinant that is increasingly recognized as a vaccine target [6,7,8].

Currently licensed vaccines for vector-borne diseases target the pathogen as is typically done with vaccines for pathogens transmissible via respiratory secretions, fecal-oral exchange, or other bodily secretions [9]. However, with the exception of yellow fever virus (YFV) and Japanese encephalitis virus (JEV), vaccine development for vector-borne diseases has been challenging. The vaccine for yellow fever virus can be considered one of the most successful vaccines given that a single immunization confers lifelong protective immunity in over 90% of vaccinated individuals [10]. YFV and JEV are the only vaccines on the WHO-approved list of vector-borne vaccines for use without caveat (Table 1). While vaccine candidates for the two most lethal mosquito-borne diseases, malaria and dengue (DENV), are not unequivocal successes. A malaria vaccine has now been made available, but requires 3 to 4 doses and yields 36% efficacy that wanes over time [11]. For DENV, a vaccine was recently approved but with the restriction that it be used only in dengue-experienced persons over the age of 9 years old in hyperendemic areas [12]. There are several candidates for vector-borne diseases including chikungunya, for which there are many Phase 1/2 candidates in the WHO pipelines [13], and leishmaniasis, a disfiguring and often lethal disease caused by the parasite *Leishmania spp*. carried by sand flies [14]. For tick-borne pathogens, there is a licensed vaccine for tick-borne encephalitis virus, but a vaccine targeting *Borrelia burgdorferi*, the causative agent of Lyme disease, has been removed from the market [15].

For the few vaccines licensed for arthropod vector-borne disease and for the majority of the candidates in the pipeline, the focus is exclusively on the pathogen. However, vaccine development for these diseases may lie at the unique interface of the hematophagous insect vector, the pathogen, and the human host (Figure 1). Notwithstanding ecological, social, and environmental determinants of health, successful transmission of vector-borne disease occurs within a triad of (1) pathogen-host interactions, (2) pathogen-vector interactions, and (3) host-vector interactions [8,15]. The opportunity for vaccine development to disrupt disease transmission at “the bite site,” where the host, pathogen, and vector initially intersect, is gaining traction [7,8,16]. Given the growing popularity of this concept, this review builds upon the existing basic science literature of cutaneous host-pathogen-vector interactions to present a broader, translational research perspective of vector-derived vaccine opportunities. Specifically, we will consider how vector delivery of a pathogen into the host skin can modulate the host immune response by focusing on three critical components: (1) the micro-environment of the bite site, (2) the local microbiome of both the vector and the host, and (3) the micro-needle for delivery of vector-borne disease vaccines into the skin.

## 2. The Micro-Environment: Why “the Bite Site” Matters

### 2.1. A Skin-Deep Immunology Review

The skin is a large complex immunoregulatory organ and functions as the main barrier tissue [17]. The skin is made up of three layers–epidermis (where the outermost layer are dead cells known as the stratum corneum), dermis, and fatty hypodermis. Each layer is complete with its own unique set of immune cells responsible for both immunosurveillance and host defense (Figure 2). Next to resident and circulating immune cells populating these tissues, epithelial cells themselves play a role in immune regulation, for example in the regulation of Th2 differentiation [18]. A detailed description of the cutaneous immune network falls outside of the scope of this review and excellent recent reviews on the topic have been published recently [19,20,21]. Once activated, the immune microenvironment facilitates rapid transport of peripheral tissue antigen via prenodal lymph and interstitial fluids to skin-draining lymph nodes so that a systemic adaptive response can be coordinated [22,23,24].

When a vector like a mosquito or fly inserts its proboscis, an elongated straw-like appendage, into the epidermis and dermis, it injects a salivary mix of active ingredients known to modulate host inflammation, hemostasis, and immunity to help the vector take a blood meal [6]. In most cases, the host immune response to the saliva is quickly dampened resulting in a small welt as a function of peripheral tolerance initiated in the skin. However, the immune response can vary as a result of an individual’s prior exposure to a vector such that pre-sensitized individuals may have delayed type hypersensitivity responses or no visible response at all [3,25,26]. This host immune downregulation extends to any antigens co-introduced at the bite site [27,28]. Without these tolerance mechanisms guided by regulatory T-cells, (Tregs), immature DCs, mast cells and basophils, vector bites—or any foreign material—would result in life-threatening situations as seen in individuals with immune dysregulation and anaphylaxis to vector bites [25,27,28,29]. During *Leishmania major* infection CD25^+^Foxp3^+^ Tregs have been shown to downregulate parasite-specific immunity, thereby allowing L. major to establish a chronic infection. This was due to CCR5-dependent homing of naturally occurring Tregs to the site of infection [25,30,31]. In the case of malaria for example, the majority of skin immunization trials require administration of mefloquine or chloroquine to induce an antitolerogenic skin microenvironment [32]. This implicates the immune microenvironment and local inflammatory status of the “bite site” in determining the outcome of infection and vaccination (Box 1).

Box 1The skinny on malaria.In the case of malaria, vaccine failure is highly correlated with the presence of sporozoites in the skin thus the majority of skin immunization trials require administration of mefloquine or chloroquine to induce an antitolerogenic skin microenvironment [32]. Other malaria immunization trials bypass the skin completely by delivering the parasite or vaccine directly into the intravascular space [33,34]. The question arises: why does natural infection via mosquito bite require only a handful of sporozoites but challenge studies require inoculation of thousands of isolated sporozoites in order to cause malaria infection [35]? To answer the question, it is necessary to consider both the tolerogenicity, or rather the host’s perceived need to dampen inflammation, versus the host immunosuppression induced by the presence of pathogen-infected vector saliva in the dermis [32].

Consider a mosquito who injects saliva infected with DENV or West Nile virus (WNV) into the extracellular spaces of the dermis as it probes for a blood vessel [36,37,38]. The host immune response begins navigating the delicate balance of inflammation versus tolerance to the mechanical intrusion of the proboscis with its accompanying saliva. Pathogens delivered via infected bite or co-inoculated with vector saliva typically cause more rapid pathogen dissemination, pathogenesis, and disease whereas pre-exposure to vector saliva or vaccination with recombinant salivary proteins reduces disease burden in animal models [39,40,41,42,43,44]. This suggests that saliva exerts local immunosuppressive effects on one hand [40,45,46,47], but that a likely rapid memory response against saliva antigens is capable of inducing a pro-inflammatory environment on the other hand, thereby contributing to the inactivation of co-administered pathogens in the epidermal-dermal microenvironment [41,42,48]. Vector salivary proteins generally appear to create a favorable immunosuppressed microenvironment for pathogen transmission and establishment of infection in the host, although exceptions exist as more novel salivary proteins and their functions are uncovered [4,40,49,50,51]. This provides an opportunity to identify vector-derived vaccination targets.

Historically, saliva peptides were thought to exclusively perturb the host cutaneous defenses via a shift from a robust Th1 response to a less effective, allergic Th2-predominant response against the pathogen in question [52]. A large body of evidence now indicates that vector salivary proteins have a myriad of impacts on immune cell function in this micro-environment [39,40,41,42,47,49,50,53,54,55]. The exact mechanisms by which vector-derived factors influence the local immune response are still being elucidated via systems biology approaches that combine transcriptomic and proteomic analyses of various vectors’ salivary glands at different developmental stages and in the presence or absence of pathogen [56]. The combination of innate and adaptive immunomodulation by vector saliva varies according to the vector, its species, and its blood-feeding strategy. For example, soft ticks feed rapidly and frequently with deep dermal penetration whereas hard ticks may feed superficially for weeks to months via specialized mouthparts [57]. Each approach elicits a different host hemostatic response and underlines the complexity and redundancy of tick saliva whose pharmacologically active properties are often species-specific [57,58]. A general overview of saliva-induced immunomodulation will be briefly reviewed here.

In the presence of mosquito saliva, the cutaneous innate immune response includes the formation of edema and the breakdown of endothelial barriers [40,49,50]. In the case of arboviruses, the presence of mosquito saliva-induced edema retains pathogen at the bite site microenvironment, facilitating ongoing arboviral infection of keratinocytes [50]. Deeper in the dermis, loss of capillary endothelial barrier integrity occurs from saliva-induced mast cell degranulation and dermal liquefaction by salivary serine proteases, contributing to disruption of immune traffic and providing the pathogen an advantage to replicate and disseminate depending upon arboviral-specific tissue tropisms [40,41,42,43,44,45,46,47,48,49].

Other early but critical innate responses of inflammation are thwarted by vector saliva. The presence of *Ixodes* and *Rhipicephalus* tick saliva reduces nitric oxide production by activated macrophages, which in turn reduces killing of pathogens like *Borrelia* bacteria [59,60,61,62]. Many tick saliva proteins from various species also inhibit human complement pathways allowing pathogens to avoid complement-dependent killing [8,46]. In mosquitos and flies, saliva predominantly immunomodulates the host’s production of IFNs and antimicrobial peptides via an increase in anti-inflammatory and Th2, or allergy-predominant, cytokine responses [45,52,63,64]. This Th2 shift can last up to 7 days after a mosquito feeding in humanized mice [47]. Decreased release of interferon-γ (IFN-γ) creates a favorable microenvironment for a pathogen, like an arbovirus or leishmaniasis, to establish infection [6]. Multiple studies have demonstrated that mosquito saliva suppresses IFNs and the subsequent cascade of IFN-stimulated gene products, allowing for establishment of arboviral infection resulting in worsened systemic disease [45,50,65,66]. On the other hand, in recent Semliki Forest Virus (SFV) infection models in immunocompetent mice, the presence of *Aedes aegypti* mosquito saliva led to chemokine expression that recruited a rapid but transient wave of neutrophils [50]. However, these neutrophils recruited myeloid cells like macrophages, monocytes and monocyte-derived dendritic cells that then served as viral targets for infection and propagation [50].

Adaptive immune response to vector saliva also impacts the success of a pathogen in establishing infection in the host. Seminal observations in the 1970s noted that rabbits exposed to tick bites were resistant to infection by tick-borne *Francisella tularensis* [4]. Research now suggests that saliva has the ability to deleteriously impact the function of both dendritic cells and macrophages, thus rendering their antigen-presenting abilities less effective. The presence of *Ae. aegypti* salivary gland extract increases dendritic cell migration to draining lymph nodes, thus hastening viral dissemination and worsening clinical disease [51,63,66,67,68]. Mosquito saliva from various species can significantly reduce T-cell lymphocyte populations via increased caspase-mediated apoptosis and dysregulation of antiviral signaling causing reduced cellular recruitment [51,63,68]. Conversely, tick saliva facilitates the pathogen’s ability to thwart effective antigen presentation by slowing the migration of dendritic cells and macrophages, thus hindering interactions with CD4^+^ T-cells [60,69].

Pathogens transmitted within vector saliva may more easily initiate host infection by taking advantage of the host’s innate and adaptive immune responses to saliva. Hence, immunization with vector salivary protein could confer protection against infection or against development of clinical symptoms. Antigen-independent mechanisms may include promotion of a more pro-inflammatory microenvironment to the pathogen, thereby minimizing its ability to establish infection in the host [6]. Humoral protection against *B. burgdorferi* infection by anti-salivary gland proteins has been well described: mice immunized with Salp15 are protected from tick-borne *B. burgdorferi* infection via an anti-Salp15 antibody-mediated mechanism that more rapidly clears Salp15-coated *B. burgdorferi* bacteria by phagocytes than in control mice [48,55]. Extrapolating these data to other possible vector-derived vaccines, plausible mechanisms of a protective humoral response could involve direct interference of anti-salivary antibodies with immunomodulatory function of salivary antigens by blocking interactions or by the generation of anti-salivary antibodies with a more pro-inflammatory profile [70]. Antibody-independent protection mediated by tissue-resident memory T cells provides protection of infection [71,72,73,74,75,76]. Indeed, the development of anti-salivary immunity via pre-exposure to vector saliva or immunization with recombinant salivary proteins can decrease disease burden in host vertebrates in a T-cell dependent manner [7,41]. In the case of *Leishmania* infection, immunization of rhesus macaques with the recombinant salivary protein PdSP15 from the sand fly *Phlebotomus duboscqi* created a Th1-driven delayed type hypersensitivity response against the parasite, increased IFN-γ expression, and reduced clinical burden of disease in cutaneous leishmaniasis [7].

The mechanisms of protection and relative contribution of each will vary with every host-vector-pathogen combination. Defining correlates of protection will be a next necessary step in vector saliva protein-based vaccine development. Given limitations of this vaccine approach, it is unlikely that localized “bite site” immunity could protect in the setting of widespread pathogen dissemination, particularly if the pathogen were introduced by an arthropod (e.g., sexual transmission ZIKV in seminal fluid) [6]. Further, the magnitude of said protection for each tripartite combination may also differ. The functional roles of newly discovered salivary and salivary gland proteins will aid in vaccine development and uncover novel mechanisms of protection that can be exploited in vaccine design.

### 2.2. Immunogen Discovery

Transcriptomic and proteomic investigations of vector salivary glands have uncovered an extraordinarily diverse and complex group of proteins [77,78]. As of 2018, transcriptomes of 49 different blood-feeding insects have been published [79]. The difference in sequencing technologies and depth of sequencing makes it difficult conclude how many of these proteins pertain directly to saliva proteins that are deposited into the host dermis. However, most fleas, mosquitos, and flies likely contain 100–200 proteins in their saliva versus ticks and kissing bugs who have more than 300 proteins in their saliva [80]. Further, nearly 40% of salivary proteins identified have no similarity to known proteins and their functions remain to be elucidated [79].

Initial proteomic analyses in the *Anopheles gambiae*, the mosquito vector of malaria, uncovered 5 saliva proteins and 122 salivary gland proteins [81]. In addition, salivary proteins in both *Anopheles* and *Aedes* mosquitos are differentially expressed in the presence of a pathogen such as DENV virus or *Plasmodium* parasites [81,82,83]. More recently, significant efforts are underway in characterizing the immunomodulatory functions of mosquito salivary proteins so that they can be appropriately targeted as prophylactics or therapeutics [84,85]. LTRIN, a 15-kDa protein in *Ae. aegypti* saliva, was identified and shown to facilitate ZIKV transmission in the host by interfering with lymphotoxin signaling and effectively disrupting the communication to lymph nodes [54]. A separate 34-kDa *Ae. aegypti* protein increases DENV replication in keratinocytes and is under development as a marker of *Aedes* vector exposure in humans [86,87]. However, functional proteomic studies with other identified salivary proteins revealed that *Ae. aegypti* collagen-binding aegyptin or certain D7 proteins increase anti-inflammatory responses or inhibit DENV replication, thus making them poor target candidates as vaccination with these proteins leads to higher host mortality [53,88,89]. The *Anopheles gambiae* salivary protein GILT (gamma interferon inducible thiol reductase), recently identified by mass spectrometry, negatively influences *Plasmodium* sporozoite movement in the mammalian host, and recombinant GILT immunization allows the establishment of an expanded liver infection by the parasite [90]. Salp15 from *Ixodes scapularis*, the vector of *Borrelia burgdorferi*, *Anaplasma phagocytophilum*, and *Babesia microti* inhibits activation of CD4 T-cells and downstream production of IL-2 via direct binding of the CD4 co-receptor [91]. While most salivary proteins appear to promote pathogen transmission and survival in the host, the roles of newly identified salivary and salivary gland proteins are discovered as functional proteomics tools are developed.

The discovery of vector microRNA (miRNA) in the host adds a new dimension to the vector-host-pathogen interaction triad, in that saliva may be modulating post-transcriptional regulation of host gene expression [92,93]. In the hard tick *Ixodes ricinus*, ten of 35 newly identified miRNAs were up to 100 times more represented in salivary glands; and via an in silico analysis of effects on the host transcriptome, a subset of these tick saliva-specific miRNAs yielded functional changes in inflammation and pain sensing [92]. Novel miRNAs were also identified in *Aedes aegypti* and *Aedes albopictus* mosquito saliva and these are differentially expressed during infection with chikungunya virus [93]. miRNAs are already being exploited as vaccine targets in neurotropic flavivirus infection via the use of miRNA response elements to attenuate live viral vaccines [94,95]. There may be an opportunity to consider the use of these targets to block the role of saliva miRNA in modulating the host immune response to enhance pathogen infection and establishment in the host.

### 2.3. Lessons Learned from Other Microenvironments

Vaccine development targeted to other epidermal-dermal microenvironments may offer insight into vector-borne diseases that are initiated in dermal tissue. Recent observations that tissue-resident memory T-cells (T_RM_) provide robust protection to infection, even in the absence of antibodies, is leading to a broader range of T-cell targets for vaccination [71,72,73,74,75,76]. T_RM_ in barrier tissues are active at the portal of pathogen entry and play a critical important role, particularly in non-lymphoid tissues like the skin or vagina where memory T-cell entry may be limited [96], and tolerance to self is high. In the skin, infection results in T_RM_-mediated global skin immunity [97,98]. Skin immunization can create both skin-resident T_RM_ and an identical population of central memory T-cells in the lymph nodes [97,98]. Vaccination at mucosal or epithelial surfaces, as opposed to intramuscular, are under investigation to generate effective T_RM_ responses to influenza and rotavirus [99,100,101].

T_RM_ reside in the periphery and proliferate locally in response to antigen, without recirculating like other memory T-cells. Hence, they are more difficult to study [102]. Zeroing in on the skin microenvironment with the aid of transcriptomics and new three-dimensional imaging modalities can identify host cell subpopulations responsible for the local and immediate protective immune memory against vector-borne components and pathogens that enter the host dermis. Given that there are twice as many T-cells in the skin than in the peripheral blood [103], it is plausible to aim for the generation of vector saliva protein-specific T_RM_ during vector-borne disease vaccination. They are strategically positioned where repeat vector challenge will occur by skin-homing receptors and are proximally located to post-capillary venules and near the dermal-epidermal interface [103]. Further, they react more quickly to cognate antigen than circulating T-cells, and both CD4 and CD8 T_RM_ are able to more rapidly produce cytokines such as IFN-γ [104].

It is not yet fully elucidated how CD4 or CD8 T_RM_ are maintained in the tissue [98,99]. We can only hypothesize that they might contribute to a tissue-resident response to vector saliva antigens, which may also vary by pathogen. CD4 T_RM_ are located deeper in the dermis, where most mosquito saliva is deposited. They are also more motile compared to CD8 T_RM_ that more resemble sessile, dendritic-like cells found in the upper layers of epidermis [71] (Figure 2). Many questions remain: can we induce the development of vector saliva protein specific T_RM_? How will they influence the immune reaction against a non-infected mosquito bite? Does T_RM_ activation change with increasing exposure to vector saliva and/or immunization with salivary proteins? Will primed skin-resident T_RM_ provide protection against vector borne pathogens? Hence, further investigations into the mechanisms of how host T_RM_ interact with vector saliva are needed.

Here, in Box 2, three other cutaneous microenvironments–melanoma, vaccinia scarification, and herpes simplex virus (HSV) infection–are briefly described in order to understand the role of T_RM_ in the development of skin tissue-targeted vaccines and consider cross-disciplinary opportunities that can be translated to vector-borne disease vaccines.

Box 2Micro-Lessons on Micro-environments in the Skin.
*Melanoma Onco-Vaccines*
Melanoma is a malignant disease that begins with the neoplastic transformation of melanocytes in the epidermis and then spreads systemically. There are several oncologic vaccine candidates for melanoma based upon extensive pre-clinical and clinical research on the cutaneous immune processes that govern the spread, or containment, of melanoma. The T_RM_ subset of tumor-infiltrating lymphocytes are a major target of immune checkpoint blockade, and their presence correlates to positive clinical outcomes [105,106]. Intradermally administered melanoma vaccines cause the development of tumor-specific T_RM_ to accumulate in both vaccinated and non-vaccinated skin [106]. The T_RM_ that are vaccine-induced, as opposed to naturally induced, suppressed tumor growth as they were able to effectively infiltrate the tumors and operate independently of circulating CD8 T-cells [107]. In lymphopenic murine hosts, tumor-specific CD4 T_RM_ had cytotoxic activity active against established melanoma tumor [108]. Clinical cancer immunotherapy via anti-melanoma vaccines targeting skin-resident T_RM_ will provide the basis for other vaccine platforms targeting cutaneous T_RM_ development for rapid response against pathogen entry.
*Vaccinia Vaccination via Skin Scarification*
Administration of vaccinia via skin scarification was the main method employed for global smallpox eradication [109]. Today, vaccinia is often used as a viral vector in vaccine development and a body of research now demonstrates that scarification, with or without the virus, leads to nonspecific immunoprotection, and importantly, the development of T_RM_ [99,109,110,111]. Delivery of vaccinia virus via superficial skin injury resulted in T-cell mediated immunity, regardless of neutralizing antibody present, and T_RM_ can target and eliminate virally infected cells shortly after viral skin inoculation challenge [111]. Similar results were also noted with skin scarification using the clinically safer poxviral vector, a nonreplicating modified vaccinia Ankara virus that is usually delivered intramuscularly [111]. In the case of mosquito-borne disease vaccines, skin scarification presents a vaccine delivery method mimicking the mosquito’s natural mode of transmitting a pathogen via intradermal probing, the phase in which it is releasing saliva and any available pathogens [112]. It is worth noting that scarification with virus, protein, or hapten can induce T-cell accumulation both at the site and globally throughout the skin [97,98], suggesting that a vector-derived immunogen delivered via scarification could also generate an effective T_RM_ population in the skin.
*HSV: The Local-Global Phenomenon of T_RM_*
Herpes simplex virus (HSV) infection causes intraepithelial vesicles in the mouth or anogenital regions and then remains latent in sensory ganglia nerves. The likelihood of HSV-2 reactivation is inversely correlated to the number of T_RM_ present in genital epithelium [113]. Specifically, CD8 T_RM_ appear after acute HSV infection and can travel between keratinocytes, similar to Langerhans cells but are not found in the stratum corneum, and occupy certain epidermal niches [71,75]. T_RM_ are an attractive vaccine target as they migrate in search of antigen, seemingly without specific attraction to infected cells, and regular killing of virally-infected cells by T_RM_ is possible, but not routinely observed in the case of HSV and CD8+ T_RM_ [75,114]. Hence, natural or vaccine-induced protection from HSV infection or reactivation requires a high density of T_RM_ serving in the role of immunosurveillance. The ability to generate sufficient HSV-specific T_RM_ density is now evident via local proliferation in response to secondary exposures to HSV without displacing existing T_RM_ [76]. Further, in the case of HSV, vaccinia virus, and Zika virus (ZIKV), sequential exposures to antigen at distinct sites of epithelial disruption leads to an increase in pathogen-specific T_RM_ throughout all uninfected skin, including distal sites like the vaginal epithelium [98,115,116].
*Vector-Borne Diseases: Another Possible Target for T_RM_?*
T_RM_ cells are crucial for local immunity and recall responses. Virus-specific skin T_RM_ appear to be long-lasting and autonomous [75]. As a parallel to repeated laboratory inoculation of a virus into the host skin to induce T_RM_ immunity, a vector repeatedly disrupts the host epidermis in daily life in an endemic area. This ongoing exposure to vector-derived antigen could possibly lead to vector-specific T_RM_ generation and replenishment throughout the skin compartment in individuals immunized with vector-derived salivary proteins. Likewise, vector protein-specific skin T_RM_ may be able to rapidly produce inflammatory cytokines such as IFN-γ, IL-2 and TNF-α upon stimulation (e.g., vector bite) anywhere in the epidermal-dermal environment. One theoretical concern of salivary-based vaccination is a heightened inflammatory reaction to naturally occurring mosquito bites, resulting in clinically significant adverse local, or possibly systemic hypersensitivity [117]. Yet, as a counterexample, a lifetime of exposure to a specific vector antigen may be protective against a specific disease. This concept is invoked in the mystery of partial *Plasmodium* immunity in longstanding residents of malaria-endemic areas [118,119] and the phenomenon of tick immunity, first noted in 1939 in guinea pigs immune to tick-borne diseases if they were previously exposed to numerous tick bites [120]. However, the question remains how a balanced and protective vector antigen-specific T_RM_ global skin immunity can be induced via vaccination.

## 3. The Microbiome: Impact of Both Host and Vector Flora on the Pathogen

The human intestinal microbiome composition drives host inflammatory and infectious disease pathogenesis as well as vaccine response, particularly for mucosal vaccines like rotavirus and polio [121,122,123,124,125]. It is plausible to consider that skin microbiomes may also drive immune response for vector-borne diseases and their associated vaccines for two reasons: (1) both the vector and vaccine deliver pathogen or antigen, respectively, via the skin and its commensal bacteria and (2) the gut microbiome of the vector can influence pathogenesis in the host skin.

Tolerizing mechanisms of skin immunoregulation, such as Tregs, are influenced by skin microbiota [126,127,128,129]. The composition of the skin microbiome can change during inflammation, but the role of the pathogen–or the vector–is not yet fully understood [130]. For example, protective immunity to the parasite *Leishmania major* transmitted by sand fly bite relies upon cutaneous T-cell dependent release of inflammatory cytokines like IFN-γ and TNF-α, both of which are greatly reduced in germ-free mice raised in aseptic conditions then challenged intradermally with *L. major* [126].

In addition to the impact of the host skin microbiome on the host immune response, the gut microbiome of the vector can drive severity of disease in the host. It is well-known that vector gut microbiomes drive vector competence, the ability of a particular vector to transmit a particular pathogen, as seen in *Ae. aegypti* mosquito midguts where the presence of *Wolbachia* dictates the ability to transmit a variety of flaviviruses [131]. Yet, the impact of the vector gut microbiome on the host-pathogen interaction in the skin was recently demonstrated. In the case of vector-transmitted *Leishmania donovani*, the causative agent of visceral leishmaniasis, the sand fly egests gut microbes into the host skin triggering an IL-1β-driven neutrophil influx [132]. Pre-treating infected sand flies with antibiotics to reduce gut microbiota impairs the inflammasome response in the host, and subsequently the parasites’ ability to establish visceral infection. However, a conflicting finding in *Anopheles gambiae* mosquitos reveals that their ability to transmit *Plasmodium falciparum* malaria is augmented when feeding on antibiotic-positive blood from a child [133]. For vector-borne vaccine deployment or testing in endemic areas with high antibiotic usage, there may be unintended consequences on both the host and vector microbiomes that impact disease transmission and vaccine efficacy.

## 4. The Micro-Needle: Rethinking Delivery

As a less painful alternative to the commonly used hypodermic needle for subcutaneous vaccine delivery, microneedles are micron-scale needles that provide a minimally invasive method for transcutaneous delivery of vaccine past the outermost layer of epidermis, the stratum corneum (SC), without activating the underlying pain receptors like conventional needles [134,135,136] (Figure 1). Available as solid, drug-coated, deep, dissolving, and hollow, microneedles may confer an advantage in vector-borne vaccine development in that they more closely mimic deposition of the pathogen or antigen into the skin microenvironment. Indeed, bioengineers have now proposed a microneedle design based on the mosquito fascicle, a collection of six stylets with serrated design and vibration specialized for painless insertion into the skin [136]. Given the initial goal of all vector-borne pathogens is to survive the skin’s initial immune assault, a vector-borne disease vaccine delivery model would also disrupt epithelium and drive antigen presentation within the pathogen’s initial target tissue, the skin. In order to develop a robust protective skin-resident response, transcutaneous vaccine delivery via microneedle may have an added advantage over the more traditional vector-borne vaccine delivered into the subcutaneous space via hypodermic needle.

Intradermal administration may also be a more optimal delivery route for vector-borne disease vaccine development. However, the risk of adverse events is higher as patients are more prone to pain, inflammation, or abscesses [135]. Microneedles present advantages over intradermal vaccine administration as they are non-invasive, causing little to no pain, and requiring little if any technical training as they can even be self-administered by patients [137]. Although microneedles may be more expensive to design, the total amount of vaccine administered via microneedle is lower than other routes and this dose-sparing benefit may offset higher costs of development [138]. Furthermore, microneedles themselves are thermostable, while coated or dissolvable microneedle eliminate the need for complicated cold-chain storage and on-site vaccine resuspension (with the exception of live, attenuated vaccines) [134,135].

Despite these advantages, microneedles have not been extensively deployed in vector-borne diseases, and more convincing clinical data is available for microneedle delivery of influenza, measles, and poliovirus vaccines [134]. For other pathogen-targeted vector-borne disease, an early published study of the “Nanopatch” microneedle described an inactivated whole chikungunya virus vaccine and a DNA-delivered WNV vaccine to demonstrate delivery of protein or DNA payloads, respectively, targeted directly to epidermal and dermal antigen-presenting cells in mice [139]. BALB/c mice vaccinated via microneedle coated with DNA plasmids of *Leishmania infantum* histones showed a Th1 bias compared to subcutaneous and intradermal routes of administration, but none were successful in controlling *Leishmania major* infection [140]. Similar to a microneedle, tattoo delivery of an arboviral SFV-based vaccine targeting human papillomavirus (HPV) resulted in increased HPV-specific cytotoxic cells and IFN-γ expression in C57BL/6 mice compared to intramuscular injection, pointing to the intrinsic immunogenic potential of intradermal delivery [141]. Part of these effects may be due to the mechanical consequences of damaged keratinocytes releasing IL-6, a pro-inflammatory cytokine, or due to inherent differences in Th1 versus Th2 responses in the chosen mouse model [67]. Regardless, an intradermal or microneedle-delivered vaccine of a vector-derived component, alone or in combination with a pathogen, may further promote a desired pro-inflammatory, Th1-driven response in the skin microenvironment.

The current major limitation of microneedle-administered vector-borne disease vaccines is a lack of clinical data across the entire infectious diseases spectrum. Of the ten registered clinical Phase 1 and 2 trials, the majority of which target influenza, patient safety and tolerability profiles published thus far are acceptable [134,142]. Still, user acceptability, scale-up from laboratory research, and dose loading capacity will be obstacles to widespread adoption of microneedle-delivered vaccines for any disease entity [135]. The role of adjuvants in microneedle-delivered vaccines for vector-borne diseases is not discussed here, but will also need to be further elucidated in terms of loading, effectiveness, and potential benefits of Th1-promoting adjuvant selection.

## 5. Conclusions

Vector-borne disease incidence continues to increase worldwide [1,2]. Innovative, broad, and integrated research efforts are needed to mitigate vector-borne diseases and their complex macroecological systems that involve climate, urbanization, and human encroachment among other factors. Vaccines are the best defense against vector-borne disease. However, it is the combined complexity of the ‘micro’—the dermal microenvironment and various microbiomes of host and vector—that challenge the development of highly effective vector-borne disease vaccines. Vaccine research may initially focus on the two most obvious ‘micro’ factors, namely the identification of vector-derived salivary components and how they influence the local immune response and the mechanical or behavioral aspects of vector-delivered pathogen entry into host skin. In order to create the next generation of successful vector-borne disease vaccines, it will be critical to elucidate the immunological cascades and key cell subpopulations in these microenvironments where vector, host, and pathogen collide.

## Figures and Tables

**Figure 1 vaccines-07-00010-f001:**
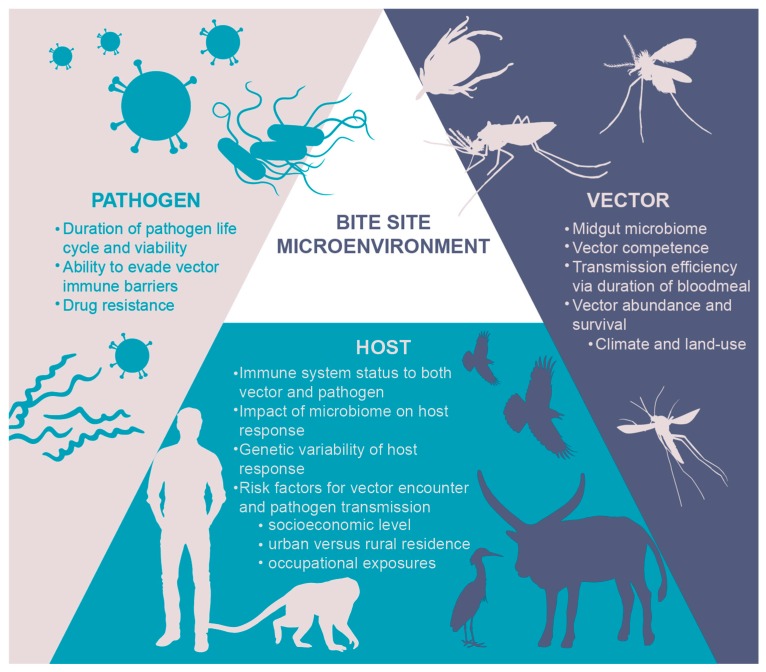
Vector-host-pathogen triad of exposure and interaction.

**Figure 2 vaccines-07-00010-f002:**
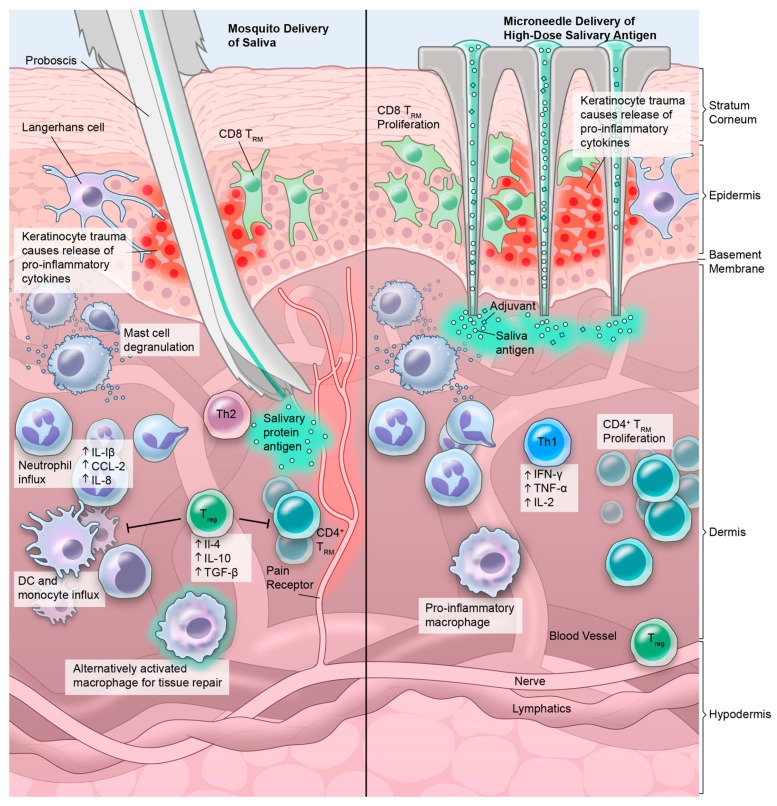
Cutaneous immune environment in the setting of mosquito saliva (left) and saliva vaccination (right). **Left panel**: The mosquito proboscis deposits saliva antigen into the dermis while also causing keratinocyte trauma and activation. Saliva antigens lead to mast cell degranulation, increase in Th2-dominant cytokines, and subsequent transient influx of neutrophils, dendritic cells and monocytes. Regulatory T-cells in the skin inhibit T cell activation and modulate antigen presenting cell function. **Right panel**: Microneedle administration of salivary antigen causes similar keratinocyte trauma and activation of cytokine release. Supratherapeutic doses of salivary antigen with Th1-promoting adjuvant are released into the dermis without activating nociceptors. A pro-inflammatory response is initiated resulting in macrophage activation and Th1 polarization. Theoretically, proliferation of salivary antigen-specific CD4^+^ tissue-resident memory (T_RM_) cells proliferation in the dermis, and possibly proliferation of CD8^+^ T_RM_ cells in the epidermis, could allow for rapid, protective T_RM_ responses to any future antigen encounter.

**Table 1 vaccines-07-00010-t001:** Status of arthropod vector-borne disease vaccines licensed or in clinical trials ^1^.

Pathogen	Primary Vector(s)	Vaccine Name	Platform	Immunogen	Adjuvant	Status ^2^	Sponsor
**Dengue virus ^2,3^ serotypes 1–4**	*Aedes aegypti* mosquito, *Aedes albopictus* mosquito	CYD-TDV	Recombinant viral vector (YFV backbone)	PrM+E of DENV1-4	None	Registered by WHO in select populations *	Sanofi-Pasteur
		TDV	Recombinant viral vector	PrM+E	None	Phase 3	Takeda
		TV003/TV005	Recombinant viral vector (DENV backbone)	Whole virus/PrM+E	None	Phase 3	NIAID
		TDENV-PIV	Inactivated whole target organism	Whole virus	Alum or AS03	Phase 2	USAMRMC
		V180	Subunit	PrM+E	Alhydrogel	Phase 1	Merck
**Zika virus ^2^**	*Aedes aegypti* mosquito	GLS-5700	DNA	PrM+E	None	Phase 1	GeneOne Life Science/Inovio
		MV-Zika	Recombinant viral vector	PrM+E	None	Phase 1	Themis Bioscience
		AGS-v	Synthetic peptide	Mosquito saliva peptide	IFA-51	Phase 1	NIAID
		mRNA-1325	mRNA	PrM+E	None	Phase 2	Moderna Therapeutics
		VRC-ZKADNA085-00-VP	DNA	PrM+E	None	Phase 1	NIAID
		VRC-ZKADNA090-00-VP	DNA	PrM+E	None	Phase 2	NIAID
		ZIKA PIV	Inactivated whole target organism	Whole virus	Alum	Phase 1	NIAID
		PIZV or TAK-426	Inactivated whole target organism	Whole virus	Alum	Phase 1	Takeda
		VLA1601	Inactivated whole target organism	Whole virus	Alum	Phase 1	Valneva Austria GmbH
**Chikungunya virus ^2^**	*Aedes aegypti* mosquito, *Aedes albopictus* mosquito	PXVX0317 CHIKV-VLP	Virus-like particle	E1, E2 and capsid proteins	With and without Alhydrogel	Phase 2	NIAID now transferred to PaxVax
		MV-CHIK	Recombinant viral vector			Phase 2	Themis Bioscience
		VAL 181388	mRNA	N.A.	N.A.	Phase 1	Moderna Therapeutics
		CHIK001 or ChAdOx1	Recombinant viral vector			Phase 1	University of Oxford
		VLA1533	Live, attenuated virus	Whole virus	None	Phase 1	Valneva SE
		BBV87	Inactivated whole target organism			Phase 1	Bharath Biotech
		CHIKV 181/25	Inactivated whole target organism			Phase 1	USAMRMC transferred to Indian Immuno-logicals
**Yellow Fever virus**	*Aedes aegypti* mosquito	Yellow Fever Vaccines (YFV) sold as YF-VAX in USA, STAMARIL elsewhere	Live, attenuated virus of 17D lineage			Licensed worldwide	Sanofi-Pastuer
**West Nile Virus**	*Culex spp*. mosquito	WN/DEN4Δ30	Recombinant viral vector	Whole live, attenuated virus		Phase 1	Johns Hopkins University
		HydroVax-001	Inactivated WNV		Alum	Phase 1	NIAID
		VRC-WNVDNA020-00-VP	DNA	PrM and E proteins of NY99 strain with CMV/R promoter		Phase 1	NIAID
**Japanese Encephalitis virus**	*Culex spp*. mosquito	CD.JEVAX^®^	Primary hamster kidney cell-derived, live, attenuated vaccine based on SA 14-14-2 strain	PrM+E		Licensed in China since 1988 as JEVAX	Chengdu Institute
		IMOJEV^®^, JE-CV^®^, ChimeriVax-JE^®^	Live, attenuated YFV with SA 14-14-2 live attenuated JEV produced in Vero cells	PrM+E		Licensed as early as 2010 in Australia and other Asian countries	Sanofi Pasteur
		Ixiaro^®^, JESPECT^®^, JEEV^®^	Inactivated Vero cell-derived	Whole virus	Alum	Licensed in USA and Europe since 2009	Valneva Austria GmbH
**Equine Encephalitis Viruses (Eastern, Western, and Venezuelan)**	*Culiseta melanura* mosquito (but human bridge vectors are likely *Aedes*, *Culex*, or *Coquillettidia spp*. mosquitos)	TSI-GSD 210, Lot 3-1-92	Inactivated WEE	Whole virus		Phase 2	USAMRMC
		TSI-GSD 104, Lot 2-1-89	Inactivated EEE	Whole virus			
		C-84, TSI-GSD 205, Lot 7	Inactivated VEE	Whole virus			
		TC-83, NDBR-102	Live, attenuated VEE				
		pWRG/VEE	DNA	pWRG/VEE		Phase 1	Ichor Medical Systems
***Plasmodium falciparum*^2^**	*Anopheles spp*. mosquito	RTS,S/ASO4 (Mosquirix^®^)	Recombinant subunit	CSP	AS01	Approved by EMA for children 6–17 months of age	Glaxo-SmithKline Inc.
		ChAd63/MVA ME-TRAP	Recombinant subunit	TRAP + ME epitopes (CS, LSA1, LSA3, STARP, EXP1, pb9)	None	Phase 2B	University of Oxford
		ChAd63/-METRAP	Recombinant subunit	ME+TRAP	None	Phase 1	University of Oxford
		ChAd63 RH5 +/− MVA RH5	Recombinant subunit	RH5	None	Phase 1	University of Oxford
		PfsSPZ	Inactivated whole organism	Whole sporozoite	None	Phase 2	Sanaria
		PfCelTOS FMP012	Recombinant subunit	CelTOS protein	AS01B or GLA-SE	Phase 1	USAMRMC
		Pfs25-EPA+Pfs230-EPA	Pfs25M or Pf230D1M conjugated to EPA, respectively	Pfs25M, Pfs230D1M	AS01 or Alhydrogel	Phase 1	NIAID
		R21	Recombinant subunit	CSP less-HepBsA	AS01B or Matrix-M1	Phase 1/2	University of Oxford
		GMZ2	Recombinant subunit	GLURP, MSP3	Aluminium hydroxide, GLA-SE	Phase 2b	Statens Serum Institute
		PRIMVAC (placental malaria)	Recombinant protein	VAR2CSA fragment	Alhydrogel or GLA-SE	Phase 1	INSERM
		SE36	Recombinant subunit	N-terminal SERA5	Alhydrogel	Phase 1	NobelPharma Co Ltd, Japan
***Plasmodium vivax***	*Anopheles spp*. mosquito	ChAd63/MVA PvDBP	Recombinant viral vector	PvDBP_RII		Phase 1	University of Oxford
***Borrelia burgdorferi***	*Ixodes scapularis*, blacklegged or deer tick	Multivalent OspA Lyme Borreliosis vaccine	Recombinant peptide	6 antigens of rOspA	Alum	Phase 1/2	Baxalta (Shire)
		VLA15	Recombinant peptide	Multivalent OspA	Alum	Phase 1	Valneva Austria GmbH
**Tick-borne Encephalitis**	Hard ticks of *Ixodidae* family	FSME-Immun (Junior)	Neudorfl strain of European subtype		Aluminum hydroxide	Licensed in Europe in 1976	
		Encepur-Adults (-Children)	K23 virus strain		Aluminum hydroxide	Licensed in Europe in 1994	
		TBE-Moscow	Sofjin strain of Far-Eastern viral subtype		Aluminum hydroxide	Licensed in Russia in 1982 (and in 1999 for children >3 years)	
		EnceVir	Far-Eastern strain 205		Aluminum hydroxide	Licensed in Russia	

^1^ Per US Centers of Disease Control and Prevention, World Health Organization Vaccine Trial Tracker for trials open and recruiting or completed updated as of May 2018, in the most recent position papers referenced in the August 2018 WHO Recommendations for Routine Immunizations, as detailed on clinicaltrials.gov, or as individually referenced; ^2^ Given some vaccine candidates have multiple trials ongoing or completed, this reflects the farthest along stage in development; ^3^ Only tetravalent dengue vaccine candidates are included; * SAGE recommendations are that this vaccine should only be given to flavivirus-experienced populations in hyperendemic areas; EMA = European Medicines Agency.

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
