# Peer review of "Time to Micromanage the Pathogen-Host-Vector Interface: Considerations for Vaccine Development"

_vaccines, 2019, doi:10.3390/vaccines7010010_

Round 1
Reviewer 1 Report
This review manuscript addresses the important topics of arthropod –host pathogen interactions, vector-borne disease suppression, and development of innovative arthropod vector directed immunization strategies. Unfortunately, significant relevant published research studies on vector-host-pathogen and vector-host immune interactions are overlooked, resulting in a far less than adequate representation of the current knowledge of the immune interactions at the host-vector-pathogen interface. Descriptions of arthropod vector host immune interactions, saliva mediators of host immune modulation, and how the arthropod creates a favorable site for pathogen establishment are superficial, fragmented, and in numerous instances inaccurate. The non-specialist reader of this submitted review would come away with an inaccurate view of host cutaneous immune defenses; the spectrum of blood feeding arthropod induced host immune responses; blood feeding arthropod modulation of host immunity; and, how these disease vectors modulate host immunity.
Lines 24-28: The introduction would leave the non-specialist reader of this review with the impression that the authors have developed the “novel” concept of anti-vector immunization against arthropod vector saliva, or other physiologically relevant molecules, as a disease control measure. Characterization of arthropod vector stimulation of host immunity and vector countermeasures have been an increasingly active and productive area of research for over four decades. The fact that host immunity to the bite of an arthropod vector can provide protection against the transmission of a highly virulent pathogen has been known since the late 1970s. The significant body of literature on anti-vector vaccines is ignored in this manuscript. The introduction to this manuscript should provide a brief accurate historical overview of the work to date citing key references. The lack of such historical perspective is all the more disappointing, since the lead author is affiliated with a laboratory whose senior investigators contributed mightily to this field over several decades.
Lines 52-53: Cite references, especially reviews that already advanced the concepts stated in these lines.
Lines 67-72: The discussion of the cutaneous immune network could be greatly enhanced by including information contained in, and citations of, excellent review manuscripts on the subject.
Pasparakis, M. et al. (2014). Nature Reviews Immunology 14, 269-301.
Nestle, F. et al. (2009). Nature Reviews Immunology 9, 679-961.
Kupper, T., Fuhlbrigge, R. (2004). Nature Reviews Immunology 4, 211-222.
Richmond, J., Harris, J. (2014). Cold Spring Harbor Perspectives in Medicine 4, a015339.
Yazdi, A. et al. (2016). Seminars in Immunopathology 38, 3-10.
Tay, S et al. (2014). Current Dermatology Reports 3, 13-22.
Since Th2 polarization by mosquitoes and ticks is discussed in this review, the recent excellent review on cutaneous Th2 responses should be cited in this manuscript.
Walker, J.A., McKenzie, A.N.J. (2018). Nature Reviews Immunology 18, 121-133.
Explanation of immune responses in skin to mosquito and tick feeding is both inaccurate in part and incomplete in regard to immune responses stimulated by blood feeding, arthropod modulation of host immunity, and creating a favorable environment for pathogen introduction and establishment. Refer to recent reviews on these topics and salivary gland transcriptomes and proteomes.
Chmelař, J. et al. (2016a). Trends in Parasitology 32, 242-254.
Chmelař, J. et al. (2016b). Trends in Parasitology 32, 368-377.
Kazimírová M, Štibrániová I. (2013). Frontiers in Cellular and Infection Microbiology 3, 43.
Kotál, J. et al. (2015). Journal of Proteomics 128, 58-68.
Wikel, S.K. (2018). Frontiers in Bioscience, Landmark 22:2105-2121.
Lines 87-88: Not all mosquito or biting fly bites induce a “small welt” in the skin. Depending on the number of bites over time, reactions can encompass type I and/or type IV immune responses and the induction of tolerance. The authors should review the following brief, but highly relevant report: Mellanby, K. (1946). Man’s reaction to mosquito bites. Nature 158, 554.
Lines 89-91: Provide a discussion, and relevant literature citations, supporting the role of regulatory T lymphocytes in modulation of host immune defenses, particularly during an initial exposure to arthropod bites.
Lines 97-98: Consider that mosquitoes are solenophages, introducing saliva directly into post-capillary venules or small pools, hematomas, created by mouthpart probing.
Lines 102-104: Further develop this theme and describe the experimental underpinnings supporting these responses.
Lines 115-120: General narrative needs to be revised to address recognized differences in host cutaneous responses to mosquito and tick bites. As written, this scheme is too simplistic and creates an inaccurate impression of the scope of host responses to blood feeding arthropods.
Lines 145-146. Authors should note the publication: Bell, F. et al. (1979). American Journal of Tropical medicine and Hygiene 28, 876-880. This manuscript reported studies in which acquired resistance to tick bite resulted in significant resistance to tick transmission of highly virulent type A Francisella tularensis.
Author Response
Dear Reviewer,
Please find our revisions per your suggestions. We detail those revisions below per each comment.
Reviewer 1:
This review manuscript addresses the important topics of arthropod –host pathogen interactions, vector-borne disease suppression, and development of innovative arthropod vector directed immunization strategies. Unfortunately, significant relevant published research studies on vector-host-pathogen and vector-host immune interactions are overlooked, resulting in a far less than adequate representation of the current knowledge of the immune interactions at the host-vector-pathogen interface. Descriptions of arthropod vector host immune interactions, saliva mediators of host immune modulation, and how the arthropod creates a favorable site for pathogen establishment are superficial, fragmented, and in numerous instances inaccurate. The non-specialist reader of this submitted review would come away with an inaccurate view of host cutaneous immune defenses; the spectrum of blood feeding arthropod induced host immune responses; blood feeding arthropod modulation of host immunity; and, how these disease vectors modulate host immunity.
Lines 24-28: The introduction would leave the non-specialist reader of this review with the impression that the authors have developed the “novel” concept of anti-vector immunization against arthropod vector saliva, or other physiologically relevant molecules, as a disease control measure. Characterization of arthropod vector stimulation of host immunity and vector countermeasures have been an increasingly active and productive area of research for over four decades. The fact that host immunity to the bite of an arthropod vector can provide protection against the transmission of a highly virulent pathogen has been known since the late 1970s. The significant body of literature on anti-vector vaccines is ignored in this manuscript. The introduction to this manuscript should provide a brief accurate historical overview of the work to date citing key references. The lack of such historical perspective is all the more disappointing, since the lead author is affiliated with a laboratory whose senior investigators contributed mightily to this field over several decades.
We thank the reviewer for his input and for remarking our lack of historical background. The introduction has been heavily revised to include the historical perspective, which was not initially included due to space constraint and the focus of how to translate this growing body of work to the bedside of the patient. It was not this author’s intention to omit the historical precedent as we have benefited greatly from the mentorship of our lab’s senior investigators who have pioneered the field and with whom we work closely.
Lines 52-53: Cite references, especially reviews that already advanced the concepts stated in these lines.
References added re: vaccine development focused on bite site (Wikal, Reed, Chmelar).
Lines 67-72: The discussion of the cutaneous immune network could be greatly enhanced by including information contained in, and citations of, excellent review manuscripts on the subject.
Pasparakis, M. et al. (2014). Nature Reviews Immunology 14, 269-301.
Nestle, F. et al. (2009). Nature Reviews Immunology 9, 679-961.
Kupper, T., Fuhlbrigge, R. (2004). Nature Reviews Immunology 4, 211-222.
Richmond, J., Harris, J. (2014). Cold Spring Harbor Perspectives in Medicine 4, a015339.
Yazdi, A. et al. (2016). Seminars in Immunopathology 38, 3-10.
Tay, S et al. (2014). Current Dermatology Reports 3, 13-22.
While there is not enough room for a detailed cutaneous immunology review, we have included these references for the readers’ benefit.
Since Th2 polarization by mosquitoes and ticks is discussed in this review, the recent excellent review on cutaneous Th2 responses should be cited in this manuscript.
Walker, J.A., McKenzie, A.N.J. (2018). Nature Reviews Immunology 18, 121-133.
Please see added at end of first paragraph of 2.1
Explanation of immune responses in skin to mosquito and tick feeding is both inaccurate in part and incomplete in regard to immune responses stimulated by blood feeding, arthropod modulation of host immunity, and creating a favorable environment for pathogen introduction and establishment. Refer to recent reviews on these topics and salivary gland transcriptomes and proteomes.
Chmelař, J. et al. (2016a). Trends in Parasitology 32, 242-254.
Chmelař, J. et al. (2016b). Trends in Parasitology 32, 368-377.
Kazimírová M, Štibrániová I. (2013). Frontiers in Cellular and Infection Microbiology 3, 43.
Kotál, J. et al. (2015). Journal of Proteomics 128, 58-68.
Wikel, S.K. (2018). Frontiers in Bioscience, Landmark 22:2105-2121.
Thank you for the additional reviews on ticks and tick-host interactions. We acknowledge that the role of tick-specific saliva may have been underrepresented given that we are more focused on the translational aspects of mosquito saliva to immunomodulate mosquito-borne diseases, but have included now more parts on the immune response to tick saliva and corresponding references now. One of the difficulties encountered in writing this manuscript was including the umbrella term of “vector saliva” as there are clear differences between vectors and how their saliva modulates the immune microenvironment. We have tried to touch upon the critical points so there is a broad understanding without inundating the “non-specialist” reader with superfluous detailed text, but we did not intend for the content to be construed as inaccurate or incomplete because we did not detail tick interactions as fully; thus we have revised with great care and added additional references that will hopefully be satisfactory to the reviewer. Given the manuscript is already lengthy from covering the required topics, Please see significant changes in Section 2 of the manuscript that details vector saliva immunomodulation.
Lines 87-88: Not all mosquito or biting fly bites induce a “small welt” in the skin. Depending on the number of bites over time, reactions can encompass type I and/or type IV immune responses and the induction of tolerance. The authors should review the following brief, but highly relevant report: Mellanby, K. (1946). Man’s reaction to mosquito bites. Nature 158, 554.
We agree with the reviewer, it is true that depending upon time and bite density, individuals may develop tolerance or DTH responses as opposed to immediate welting of the skin. We are familiar with the Mellanby report. We have included this caveat to clarify the diversity of response.
Lines 89-91: Provide a discussion, and relevant literature citations, supporting the role of regulatory T lymphocytes in modulation of host immune defenses, particularly during an initial exposure to arthropod bites.
We have corrected that the mechanisms of immune tolerance to salivary proteins is guided not only by Tregs, but also by mast cells, basophils and immature dendritic cells, as mentioned in the referenced papers (17-20). We have added the studied mechanisms of Treg suppression to L. major infection where Tregs have been shown to downregulate parasite‐specific immunity, thereby allowing L. major to establish a chronic infection (References 25, 30-31).
Lines 97-98: Consider that mosquitoes are solenophages, introducing saliva directly into post-capillary venules or small pools, hematomas, created by mouthpart probing.
We acknowledge that ultimately they introduce saliva when taking a blood meal, but also while searching for a blood meal such that extracellular spaces was used in this description given the mosquito is depositing saliva as they probe prior to successfully finding a vessel or creating a hematoma.
Lines 102-104: Further develop this theme and describe the experimental underpinnings supporting these responses.
This sentence is a general introduction to the non-specialist reader that saliva creates local immunosuppression. Further mechanistic insight and details on cutaneous innate and inflammatory immunomodulation by saliva is provided in the paragraphs below (line 121-442). We have removed the phrase “tissue-resident” given lack of direct evidence, although this is hypothesized given the memory response is likely rapid in order to control the infection in the bite site (before spread to draining lymph nodes).
Lines 115-120: General narrative needs to be revised to address recognized differences in host cutaneous responses to mosquito and tick bites. As written, this scheme is too simplistic and creates an inaccurate impression of the scope of host responses to blood feeding arthropods.
Please see extensive changes to this section that lay out the species-specific immunomodulation by ticks and the differences in which ticks feed from mosquitos. For the non-specialist Vaccines reader, we are hoping to introduce the concept of salivary immunomodulation and a general overview. While we do not aim to be “too simplistic,” we also feel that there is not enough space to detail each vector (although we have added additional tick references) and each vector species-specific immunomodulatory and blood feeding habits so have focused on the main points of immunomodulation that are becoming the focus of vaccine development.
Lines 145-146. Authors should note the publication: Bell, F. et al. (1979). American Journal of Tropical medicine and Hygiene 28, 876-880. This manuscript reported studies in which acquired resistance to tick bite resulted in significant resistance to tick transmission of highly virulent type A Francisella tularensis.
Noted and included with reference in introduction paragraph as well.
Reviewer 2 Report
This is a very well written review that summarizes potential options for vaccine development for vector-borne diseases.
I only have a few suggestions for the figures.
Figure 1. Instead of using a triangle, is it more reasonable to use "Venn diagram" to demonstrate the relations among pathogens, host, and the vector? Three circles would represent pathogen, host, and vector, respectively, the overlapping region in the center would be "bite site microenvironment".
Figure 2. A title sentence that summarizes the entire figure is needed.
Author Response
Thank you for your comments. We have made extensive edits per other reviewers' comments. For your comments, we detail below.
Figure 1. Instead of using a triangle, is it more reasonable to use "Venn diagram" to demonstrate the relations among pathogens, host, and the vector? Three circles would represent pathogen, host, and vector, respectively, the overlapping region in the center would be "bite site microenvironment".
Response: Excellent suggestion and I agree as the initial figure that we drew by hand was a Venn diagram. However, ultimately based upon the ancillary graphics included, the professional illustrator suggested the triangle format, which we think effectively portrays the message so we will respectfully decline reverting back to the Venn diagram.
Figure 2. A title sentence that summarizes the entire figure is needed.
Response: Added title as Cutaneous immune environment in the setting of mosquito saliva (left) and saliva vaccination (right).
Reviewer 3 Report
The manuscript by Manning and Cantaert reviews the role of the pathogen-host-interface in arthropod-transmitted infection and vaccination. The manuscript is very well-written and expertly covers a fascinating topic. Only minor revisions are suggested.
Minor concerns:
1) Line 71, remove "are rapidly"
2) Figure 2, define "Trm" in the legend, as it is not discussed until later in the review
3) Line 155, insert "and" between "proteins" and "their"
4) Line 194, insert "be" between "may" and "limited"
5) Line 204, Trm are unlikely targets for vaccination, since they are memory cells. Please clarify.
Author Response
Thank you for your comments. We have made extensive edits but to your comments below, we detail them here:
The manuscript by Manning and Cantaert reviews the role of the pathogen-host-interface in arthropod-transmitted infection and vaccination. The manuscript is very well-written and expertly covers a fascinating topic. Only minor revisions are suggested.
Minor concerns:
1) Line 71, remove "are rapidly" - REMOVED
2) Figure 2, define "Trm" in the legend, as it is not discussed until later in the review - DEFINED
3) Line 155, insert "and" between "proteins" and "their" - ADDED
4) Line 194, insert "be" between "may" and "limited" - INSERTED
5) Line 204, Trm are unlikely targets for vaccination, since they are memory cells. Please clarify. – CLARIFIED AS Given that there are twice as many T-cells in the skin than in the peripheral blood [84], it is plausible to aim for the generation of vector saliva protein-specific TRM during vector-borne disease vaccination.
Reviewer 4 Report
Collectively, the manuscript entitled, “Time to Micromanage the Pathogen-Host-Vector Interface: Consideration for Vaccine Development”, provides a landscape of the acute and localized events associated with early “bite-site” reactions. The authors further offer insight for future focus of vector-borne pathogens with particular emphasis on very intriguing vector salivary proteins and targeted vaccine strategies to mimic natural route of exposures. The graphics provided were well illustrated and aligned with test. I feel that this review would be of value to the scientific community and warrants publication in Vaccines. However, the below comments and revisions should be considered prior to acceptance.
Major/Minor Comments:
1) Overall the manuscript suffered from several grammatical and sentence structure errors which distracted the reader. Examples: Lines 35, 36, 70-71, 90,155,175, etc. In addition, there is a lack of consistency with use and choice of abbreviation. As examples Dengue and West Nile Virus was called to multiple times without abbreviation. Significant editing should be considered.
2) The authors negate several important aspects in their vaccine design consideration. Most notably the identification of correlates of protection (e.g, cellular or humoral). In the current form and particularly Figure 2, there is specific focus on tissue-resident T-cells. However, as identified in the Salp15 examples (Refs 35,64) antibody responses are likely the mode of protection in this B. burgdorferi model. Given that the authors propose potential targeting of salivary proteins, they should propose immune mechanisms of protection. It is this reviewer’s opinion that a robust antibody response would be requisite. It is not clear how the low amount of natural salivary proteins would provide capability to target them effectively. In the case of classical viral infection, the high levels of say viral glycoproteins are amplified in the host and provides attractive target for vaccines.
3) Along with the above comment, Figure 2 would suggest that tissue resident T-cells are critical which isn’t fully appreciated and lacks supporting data. Further, the vague call to adjuvants to promote the skewed Th1 response is misleading. Do the authors suggest that future vector-borne vaccines would be more effective with the use of Th1 skewing adjuvants such as PRR ligands? In addition, do the authors propose that bite-site tissue resident immunity will provide protection even with pathogen dissemination? Finally, immunogenicity will greatly vary with microneedle vaccination depending on the vaccine modality (e.g, protein, DNA), adjuvant (e.g., Alumn, MPLA, PolyIC) and model (e.g., Th1 vs Th2 differences in commonly used mouse strains C57BL/6 and Balb/C). The authors should clarify as to not mislead the readers.
Author Response
Thank you for your comments. We have made extensive edits. Detailed replies to your comments are here:
Collectively, the manuscript entitled, “Time to Micromanage the Pathogen-Host-Vector Interface: Consideration for Vaccine Development”, provides a landscape of the acute and localized events associated with early “bite-site” reactions. The authors further offer insight for future focus of vector-borne pathogens with particular emphasis on very intriguing vector salivary proteins and targeted vaccine strategies to mimic natural route of exposures. The graphics provided were well illustrated and aligned with test. I feel that this review would be of value to the scientific community and warrants publication in Vaccines. However, the below comments and revisions should be considered prior to acceptance.
Major/Minor Comments:
1) Overall the manuscript suffered from several grammatical and sentence structure errors which distracted the reader. Examples: Lines 35, 36, 70-71, 90,155,175, etc. In addition, there is a lack of consistency with use and choice of abbreviation. As examples Dengue and West Nile Virus was called to multiple times without abbreviation. Significant editing should be considered.
Response: Thank you for the remark. We have corrected viral abbreviations and made global editing changes.
2) The authors negate several important aspects in their vaccine design consideration. Most notably the identification of correlates of protection (e.g, cellular or humoral). In the current form and particularly Figure 2, there is specific focus on tissue-resident T-cells. However, as identified in the Salp15 examples (Refs 35,64) antibody responses are likely the mode of protection in this B. burgdorferi model. Given that the authors propose potential targeting of salivary proteins, they should propose immune mechanisms of protection. It is this reviewer’s opinion that a robust antibody response would be requisite. It is not clear how the low amount of natural salivary proteins would provide capability to target them effectively. In the case of classical viral infection, the high levels of say viral glycoproteins are amplified in the host and provides attractive target for vaccines.
Please see significant revision of the last paragraph in 2.1 addressing mechanisms. While B. burgdorferi-Ixodes tick saliva model is well-defined as antibody-mediated protection, research into other vector saliva-host-pathogen models are still lacking clearly defined antibody-mediated mechanisms of protection. Salivary proteins undergo post-trans modifications of glycosylation but likely not amplified in the host to create a target as in the case of viral glycoproteins so this is not discussed.
3) Along with the above comment, Figure 2 would suggest that tissue resident T-cells are critical which isn’t fully appreciated and lacks supporting data. Further, the vague call to adjuvants to promote the skewed Th1 response is misleading. Do the authors suggest that future vector-borne vaccines would be more effective with the use of Th1 skewing adjuvants such as PRR ligands? In addition, do the authors propose that bite-site tissue resident immunity will provide protection even with pathogen dissemination? Finally, immunogenicity will greatly vary with microneedle vaccination depending on the vaccine modality (e.g, protein, DNA), adjuvant (e.g., Alumn, MPLA, PolyIC) and model (e.g., Th1 vs Th2 differences in commonly used mouse strains C57BL/6 and Balb/C). The authors should clarify as to not mislead the readers.
Yes, we do not include an in-depth discussion of adjuvants in this review. The reviewer is correct that the authors believe Th1 skewing adjuvants would make vector-borne vaccines more effective and have included revisions for clarification (line 146, line 797). We have also added the limitation of saliva-based vaccines in that pathogen dissemination would like overpower bite site immunity and lead to disease (line 438-440). Lastly, the reviewer is correct that there are numerous possibilities of vaccine/microneedle/adjuvant combinations that are not all listed here and caveats were added to text (new lines 786-797).
Round 2
Reviewer 1 Report
The revised manuscript is acceptable for publication in the current form.
Author Response
Thank you for acceptance of our revisions. We appreciate your comments that greatly improved the manuscript.
Reviewer 4 Report
The reviewer appreciates the authors’ attention to the provided comments and suggestions. However, while the current revision has been significantly improved with respect to editing, it is still suffers in the conceptualization of vaccine design called to during the initial review.
Specifically, the authors’ generalization on saliva protein-mediated Th1 responses for protection and the subsequent focus on vaccine design is not fully substantiated. The authors’ statement in lines 121-124 support that this historical view has been abandoned in-light of “a large body of evidence now indicates…..a myriad of impacts on immune cell function…(ref 36-39,44,46,47,50-52). As such, the authors’ focus, detailed illustration, and generalization towards vaccine design is unclear.
In addition, the authors’ considerations for vaccine design particularly associated with microneedle delivery is not fully developed and is misleading to the reader. However, the authors’ do nicely provide new text (lines 165-188) on the essentials of defining correlates of protection. Nonetheless, given the authors’ choice to focus on tissue-resident immunity it would have been more fitting to discuss how adaptive responses in lymphoid structures result in the generation of skin tissue-resident T cells. And then, how alternative vaccination strategies or routes of administration may be able to skew central and peripheral memory, using say the microneedle delivery as an example. This is of particular importance given key differences in the distribution and subsets of antigen presenting cells (e.g., Langerhans DCs, plasmacytoid DCs, inflammatory macrophages, etc.) within the locations of vaccination (e.g., epidermis, dermis, muscle) and how differing APCs shape the adaptive responses. While the authors do try to suggest that microneedle delivery may be more suitable for vector-borne vaccine design, they do not provide this connection which is commonly considered in all vaccine design.
Moreover, the authors’ interpretation and comments on the advantages of microneedle delivery is misleading in several cases (lines 317-340). In particular, (lines 321-322) the authors state “In other pathogen-targeted vector-borne disease vaccine candidates, microneedle-administered adenoviral-based vaccines for ZIKV and malaria generated greater and protective immunity and antigen-specific antibodies, respectively, in C57BL/6 mice [121,122]”. This statement is confusing as to “greater” to what (differing routes or strategies??) and the references do not support this. Specially, the Zika study referenced (122) does not demonstrate competitive advantages of microneedle delivery. In this particular report, they tested an Adeno-based vaccine given subcutaneous and a recombinant protein-based microneedle vaccine given ID. Given the current differences in antigen source and the fact that the adenoviral vaccine given subcutaneous resulted in enhanced immunogenicity and protection compared to the microneedle vaccine, it is unclear how this supports the authors’ claims. This is also true with the authors’ view of results from reference 121. The study cited was limited to only an adenoviral vaccine with long-standing issues associated with boosting and vector-immunity. Nonetheless, work within reference 121 found that a combination of adenoviral and microneedle heterologous prime-boost vaccine strategies was superior to microneedle vaccination alone.
Given the above concerns, this reviewer does not feel that the current manuscript would offer benefits to researchers within this specific field or provide guidance for future vaccine development against vector-borne pathogens.
Author Response
We appreciate the reviewer's comments and have made clarifications that we hope are more satisfactory.
We have made edits, in particular to remove references 121 and 122 plus our conclusions given that the reviewer felt that these were misleading.
We felt given space limitations we were not able to address all the additional points on microneedle administration of vaccines that are highlighted by the reviewer.